# Embryonic origins of forebrain oligodendrocytes revisited by combinatorial genetic fate mapping

Yuqi Cai[†], Zhirong Zhao[†], Mingyue Shi[†], Mingfang Zheng, Ling Gong, Miao He*

Institutes of Brain Science, State Key Laboratory of Medical Neurobiology and MOE Frontiers Center for Brain Science, Department of Neurobiology, Zhongshan Hospital, Fudan University, Shanghai, China

*For correspondence:
hem@fudan.edu.cn

[†]These authors contributed equally to this work

Competing interest: The authors declare that no competing interests exist.

**Abstract** Multiple embryonic origins give rise to forebrain oligodendrocytes (OLs), yet controversies and uncertainty exist regarding their differential contributions. We established intersectional and subtractional strategies to genetically fate map OLs produced by medial ganglionic eminence/preoptic area (MGE/POA), lateral/caudal ganglionic eminences (LGE/CGE), and dorsal pallium in the mouse brain. We found that, contrary to the canonical view, LGE/CGE-derived OLs make minimum contributions to the neocortex and corpus callosum, but dominate piriform cortex and anterior commissure. Additionally, MGE/POA-derived OLs, instead of being entirely eliminated, make small but sustained contribution to cortex with a distribution pattern distinctive from those derived from the dorsal origin. Our study provides a revised and more comprehensive view of cortical and white matter OL origins, and established valuable new tools and strategies for future OL studies.

## eLife assessment

In this study the authors revisited the question of the embryonic origin of telencephalic oligodendrocytes using some new and powerful genetic tools. There is **convincing** evidence to support previous suggestions of a predominantly cortical origin of oligodendrocytes in the cerebral cortex, however the new studies suggest that LGE/CGE-derived oligodendrocytes make a modest contribution in some areas, while MGE/POA-derived oligodendrocytes make a small but enduring contribution. The findings are **valuable** and should be of interest to developmental and myelin biologists.

## Introduction

Oligodendrocytes (OLs) are an important class of macroglia responsible for producing the myelin sheaths that insulate and protect neuronal axons. Forebrain OLs arise from multiple embryonic origins. Previous fate-mapping study using *Nkx2.1^Cre* (*Xu et al., 2008*), *Gsh2^Cre* (*Kessaris et al., 2006*), and *Emx1^Cre* (*Gorski et al., 2002*) reported consecutive and competing waves of OLs derived from medial ganglionic eminence/preoptic area (MGE/POA), lateral/caudal ganglionic eminences (LGE/CGE), and dorsal pallium (*Kessaris et al., 2006*). The first wave of OLs generated by MGE/POA (MPOLs) was believed to be eliminated postnatally, while those from the second and third waves (LCOLs and dOLs) survive and populate the cortex and corpus callosum at comparable proportions. Several other studies provided both supporting and contradicting evidence to this model (*Nakahira et al., 2006*; *Tsoa et al., 2014*; *Naruse et al., 2016*; *Orduz et al., 2019*; *Liu et al., 2021*; *Shen et al., 2021*; *Tripathi et al., 2011*; *Winkler et al., 2018*). Moreover, *Gsh2* was recently found to be expressed in dorsal progenitors (*Zhang et al., 2020*), casting doubt on the interpretation of lineage tracing data from *Gsh2^Cre*.

In this study, we generated new genetic tools and combinatorial fate-mapping strategies which allow direct visualization and comparison among OLs derived from different origins. We found that neocortical OLs are primarily composed of dOLs, rather than similar proportions of $_{LC}$OLs and dOLs. In contrast, $_{LC}$OLs and dOLs made comparable contributions to piriform cortex. We also found that although $_{MP}$OLs only make a small contribution, they do persist in the cortex beyond adulthood with a unique spatial pattern distinct from that of the dOLs. In the two major white matter commissure tracts, dOLs are the vast majority in corpus callosum but make little contribution to anterior commissure, while $_{LC}$OLs behaved the opposite. These findings significantly revised the classical view and provided a new and more comprehensive picture of cortical and white matter OL origins.

## Results and discussion

To unambiguously track OLs from different embryonic origins, we first generated a knock-in driver, *Opalin^{P2A-Flpo-T2A-tTA2}* (*Figure 1*), orthogonal to Cre drivers that label dorsal or ventral progenitors (*Progenitor^{Cre}*). *Opalin* (also known as *Tmem10*) encodes oligodendrocytic myelin paranodal and inner loop protein that are specifically expressed in differentiated OLs (*Kippert et al., 2008*; *Yoshikawa et al., 2008*; *Jiang et al., 2013*; *Marques et al., 2016*). In *Opalin^{P2A-Flpo-T2A-tTA2}*, Flpo and tTA2 were inserted before the STOP codon and linked by self-cleavage peptide P2A and T2A (*Figure 1A–C*), allowing co-transcription and translation with *Opalin*. Flp-mediated recombination by this driver (hereinafter referred to as *Opalin^{Flp}* for simplicity) enables highly specific, efficient, and irreversible OL labeling, while the tTA2 component offers the flexibility for OL-specific labeling in tunable densities (*Figure 1D–F*).

Next, we established two types of genetic combinatorial fate-mapping strategies to directly visualize OLs from different embryonic origins (*Figure 2*): (1) combining *Opalin^{Flp}* and *Progenitor^{Cre}* with intersectional reporters Ai65 to label OLs derived from Cre+ progenitor domain by RFP (*Figure 2A*); (2) combining *Opalin^{Flp}* and *Progenitor^{Cre}* with RC::FLTG (*Plummer et al., 2015*) to simultaneously label OLs derived from Cre+ progenitors by green fluorescent protein (GFP) and OLs derived from the complementing Cre− progenitors by RFP (*Figure 2B*, *Figure 2—figure supplement 1A*). The first approach allowed us to track dOLs and $_{MP}$OLs (*Figure 2C, E*). The second approach empowered us to observe and compare OLs generated from dorsal and ventral origins (*Figure 2—figure supplement 1B*), or those from Gsh2+ and Gsh2− progenitors, in the same brain (*Figure 2—figure supplement 1C*). Importantly, the subtraction power enabled us to target OLs derived from LGE/CGE progenitors that express neither *Emx1* nor *Nkx2.1* (*Figure 2D* and *Figure 2—figure supplement 1D*). In addition, these strategies greatly facilitated the identification of OLs derived from specific origin which exist at relatively low density in certain regions.

Deploying these strategies, we assessed the differential contributions of dOLs, $_{LC}$OLs, and $_{MP}$OLs by analyzing RFP+ cells in the following mice: *Opalin^{Flp}::Emx1^{Cre}::Ai65* (*Figure 2C*), *Opalin^{Flp}::Emx-1^{Cre}::Nkx2.1^{Cre}::RC::FLTG* (*Figure 2D*), and *Opalin^{Flp}::Nkx2.1^{Cre}::Ai65* (*Figure 2E*). To better assess their contributions to the total OL population (*Figure 2F*), we co-stained RFP with the mature OL marker aspartoacylase (ASPA) (*Huang et al., 2023*; *Figure 2G–I*) and quantified the ratio of co-localization (*Figure 2J*). Notably, all RFP+ cells are ASPA+, reassured the specificity of our label strategies. We observed two significant differences from the traditional model in the neocortex. The first major deviation is that, instead of comparable contributions by dOLs and $_{LC}$OLs, the vast majority of neocortical OLs were dOLs but not $_{LC}$OLs. The densities (*Figure 2G–I* and *Figure 2—figure supplement 2A–F*) and ASPA ratios (*Figure 2J*) of dOLs are much higher than those of $_{LC}$OLs. Considering the possibility of incomplete recombination in combinatorial reporters, and the relatively low Cre activity in the dorsal MGE of *Nkx2.1^{Cre}* (*Xu et al., 2008*), the genuine contribution of $_{LC}$OLs to the neocortex could be even lesser than our current observation. Therefore, the large quantity of neocortical OLs labeled by *Gsh2^{Cre}* in previous study (*Kessaris et al., 2006*) or by GFP in *Opalin^{Flp}::Gsh2^{Cre}::RC::FLTG* (*Figure 2—figure supplement 1C*) most likely were predominantly dOLs generated by Gsh2+ dorsal progenitors (*Zhang et al., 2020*), rather than bona fide $_{LC}$OLs.

The second major deviation is that cortical $_{MP}$OLs are not completely depleted postnatally. Instead, they make a small but continued contribution with a unique spatial distribution pattern (*Figure 2I, J* and *Figure 2—figure supplement 2D–G*). $_{MP}$OLs display a clear rostrocaudal density decline (*Figure 2—figure supplement 2E, F*), a higher density in somatosensory cortex (SS) than motor cortex (Mo) (*Figure 2I*), and a laminar preference toward layer 4 (L4) in SS (*Figure 2—figure supplement 2G*).

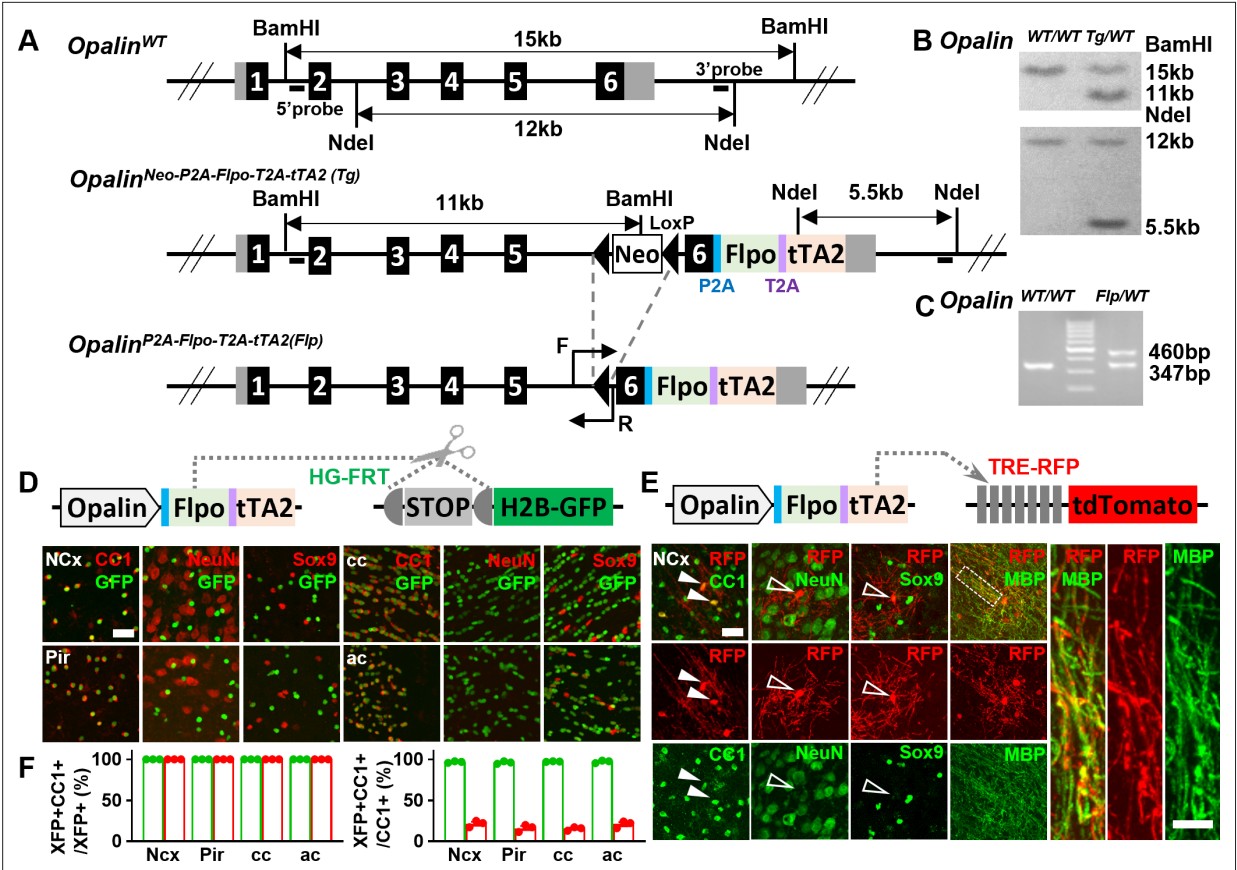

**Figure 1.** A new driver mouse for efficient and specific oligodendrocyte (OL) labeling. (**A**) Scheme for generating the *Opalin^P2A-Flpo-T2A-tTA2* allele. (**B**) Southern blot confirmation of correctly targeted embryonic stem cell clone. (**C**) Genomic polymerase chain reaction (PCR) to genotype F1 offspring. (**D**) OL labeling by Flp. (**E**) OL labeling by tTA2. High magnification images of the boxed region showing co-localization of red fluorescent protein (RFP) with myelin basic protein (MBP) staining, which further demonstrated the myelination ability of labeled OLs. (**F**) Quantification of labeling specificity (left panel) and efficiency (right panel) by colocalization with OL marker CC1. Both reporting systems are highly specific, as shown by the complete co-localization of fluorescent protein (XFP) with OL marker (CC1) and lack of co-staining with neuronal marker (NeuN) or astrocyte marker (Sox9). Quantification bar graph was not presented for NeuN and Sox9 as zero co-localizations were observed in all analyzed regions. Close to complete OL labeling was achieved by Flp-dependent H2B-GFP reporter in all analyzed regions (green dots), while sparser labeling with variable regional density was achieved by tTA2-dependent tdTomato reporter driven by TRE promoter (red dots). NCx: neocortex. Pir: piriform cortex. cc: corpus callosum. ac: anterior commissure. Scale bar: 50 μm in low magnification images, 5 μm in high magnification images. Quantification: $n = 3$. Dots represent data from individual mice.

The online version of this article includes the following source data for figure 1:

**Source data 1.** Raw unedited blot for *Figure 1B*.

**Source data 2.** Uncropped and labeled blot for *Figure 1B*.

**Source data 3.** Raw unedited gel for *Figure 1C*.

**Source data 4.** Uncropped and labeled gel for *Figure 1C*.

**Source data 5.** The raw data for the visualization of data presented in *Figure 1F*.

In contrast, the distribution of dOLs and ₗ𝒸OLs do not vary significantly across the rostrocaudal axis (*Figure 2—figure supplement 2E, F*) or between Mo and SS (*Figure 2G, H*), but exhibits increased density toward deeper layers (*Figure 2—figure supplement 2G*). Importantly, we have observed cortical ₘₚOLs in mice as old as 1 year (*Figure 2—figure supplement 2H*), well beyond the age analyzed in previous reports (*Kessaris et al., 2006*; *Orduz et al., 2019*; *Liu et al., 2021*), suggesting a persisted contribution.

We then turned our attention to the lateral three-layer archicortex, piriform cortex (Pir). Different from the neocortex, Pir contains higher proportions (*Figure 2J*) of ₗ𝒸OLs than dOLs. ₘₚOLs make the lowest contribution (*Figure 2J*) at a density similar to SS and higher than Mo (*Figure 2I*).

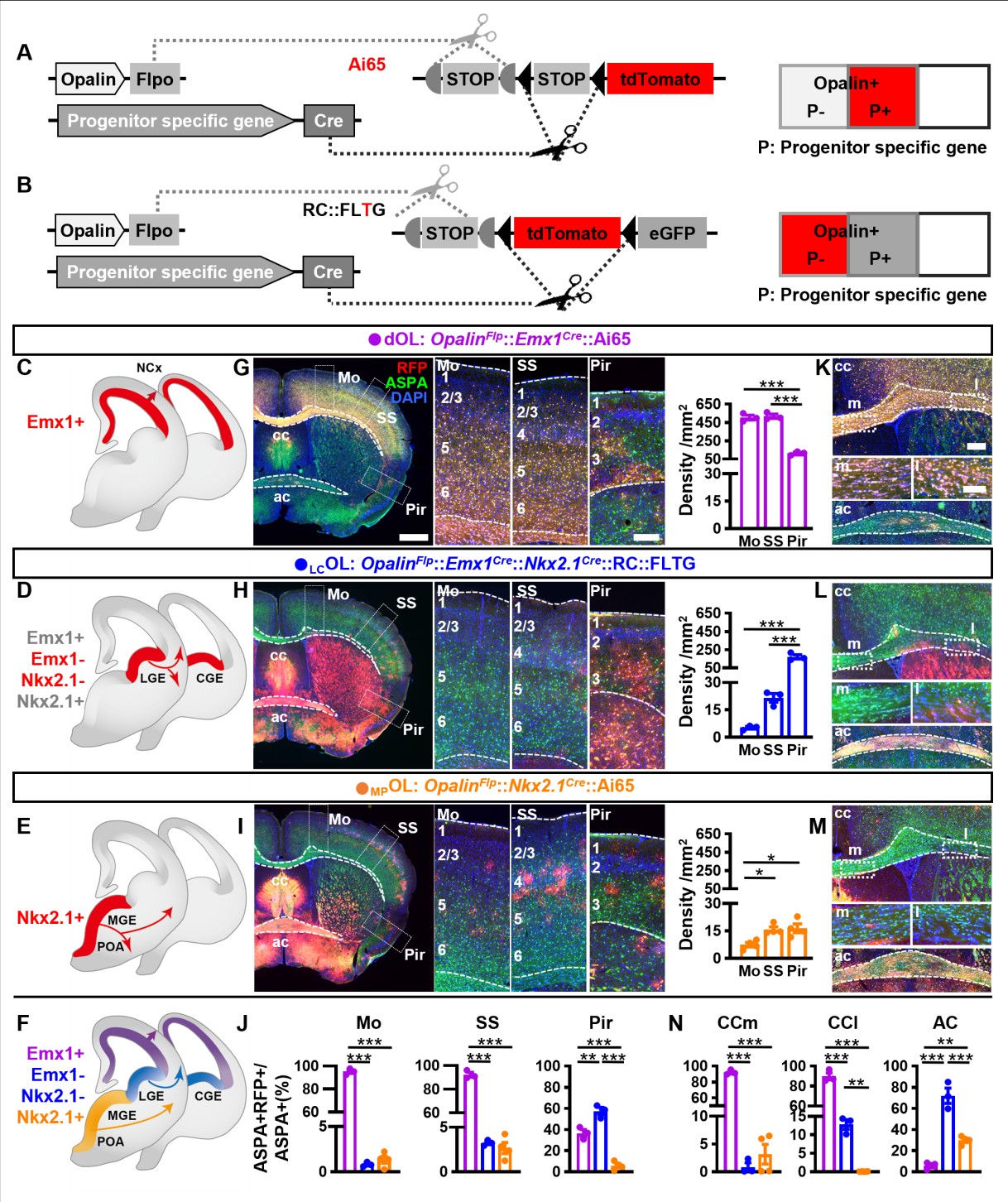

**Figure 2.** Combinatorial fate mapping of dOLs, MPOLs, and LCOLs. (**A**) Strategy for intersectional labeling. Flp-AND-Cre labels oligodendrocytes (OLs) from Cre-expressing progenitors with RFP. (**B**) Strategy for subtractional labeling of OLs derived from non-Cre-expressing progenitors with RFP. The eGFP expressing OLs derived from Cre-expressing progenitors were not used for analysis in this scenario and thereby were not highlighted by color. Schematics showing intersectional labeling of dOLs in *Opalin*$^{Flp}$::*Emx1*$^{Cre}$::Ai65 (**C**), subtractional labeling of LCOLs in *Opalin*$^{Flp}$::*Emx1*$^{Cre}$::*Nkx2.1*$^{Cre}$::RC::FLTG (**D**), intersectional labeling of MPOLs in *Opalin*$^{Flp}$::*Nkx2.1*$^{Cre}$::Ai65 (**E**), and cortical OLs derived from all three origins (**F**). (**G–I**) Representative images (left panels) and quantifications (right panels) of RFP+ cell density in motor cortex (Mo), somatosensory cortex (SS), and piriform cortex (Pir). (**J**) Quantification of differential contribution to ASPA+ OLs by three embryonic origins to Mo, SS, and Pir. Representative images (**K–M**) and quantifications (**N**) of differential contribution to ASPA+ OLs by three embryonic origins in the two major commissure white matter tracts: corpus callosum (cc) and anterior commissure (ac). MPOLs and LCOLs preferentially reside in the medial and lateral cc (cc-m and cc-l), respectively.

*Figure 2 continued on next page*

*Figure 2 continued*

Scale bar: 1 mm in low magnification images in (**G–I**), 250 µm in high magnification images of the boxed area in (**G–I**) and low magnification images in (**K–M**), 100 µm in high magnification images of the boxed area (cc-m and cc-l) in (**K–M**). n = 3 for dOLs and ₗᴄOLs; n = 4 for ₘₚOLs. Dots represent data from individual mice. Error bar: standard error of the mean (SEM). *p < 0.05, **p < 0.01, ***p < 0.001.

The online version of this article includes the following source data and figure supplement(s) for figure 2:

**Source data 1.** The raw data for the visualization of data presented in *Figure 2G–J, N*.

**Figure supplement 1.** Simultaneous differential labeling of oligodendrocytes (OLs) derived from complementary embryonic origins.

**Figure supplement 2.** The distribution pattern of cortical dOLs, ₘₚOLs, and ₗᴄOLs.

**Figure supplement 2—source data 1.** The raw data for the visualization of data presented in *Figure 2—figure supplement 2F, G*.

**Figure supplement 3.** Intersectional labeling of oligodendrocytes (OLs) derived from both dorsal origin and medial ganglionic eminence/preoptic area (MGE/POA).

These combinatorial models also grant us the opportunity to revisit the differential contributions of dOLs, ₗᴄOLs, and ₘₚOLs to the two commissural white matter tracts, corpus callosum (cc), and anterior commissure (ac), which contain high density of OLs (*Figure 2K–M*). We found that, similar to the neocortex, cc is mainly populated by dOLs and supplemented by very low proportions of ₗᴄOLs and ₘₚOLs (*Figure 2K–N*). Interestingly, ₗᴄOLs and ₘₚOLs seem to show preferential distribution in the lateral and medial regions of cc (cc-l and cc-m), respectively (*Figure 2L–N*). Different from

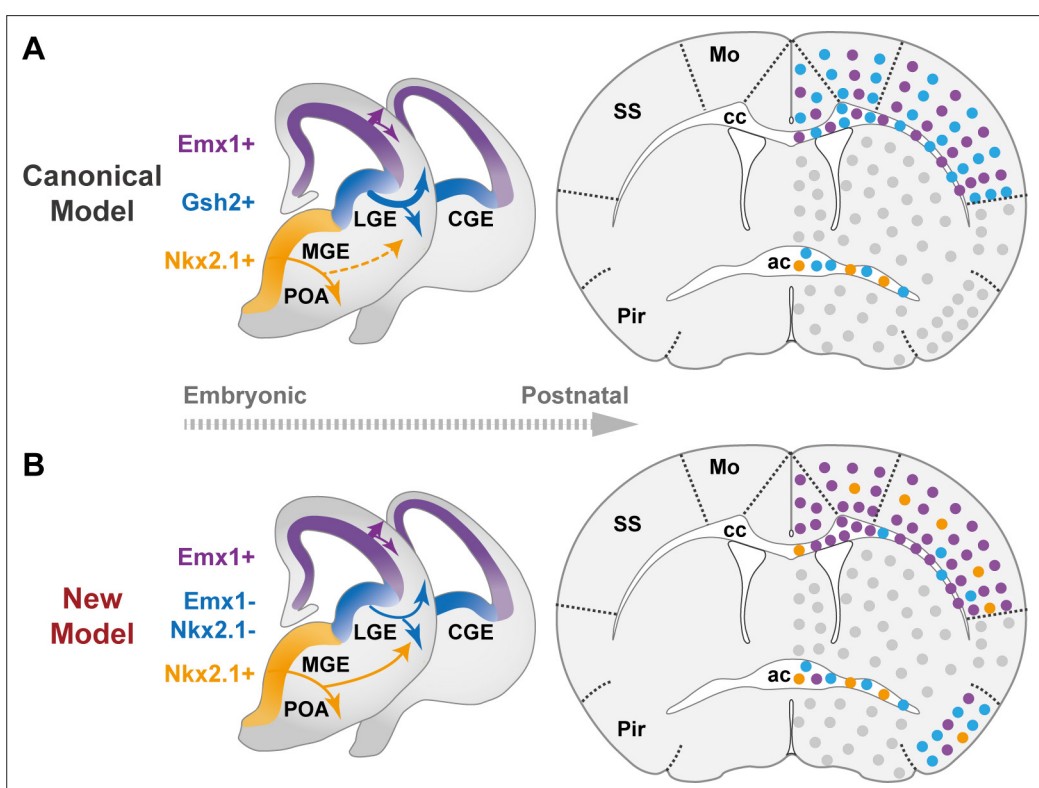

**Figure 3.** The classical and revised model of forebrain oligodendrocyte (OL) origins. (**A**) In the classical model (*Kessaris et al., 2006*), OLs derived from medial ganglionic eminence/preoptic area (MGE/POA) (orange) were largely eliminated postnatally (thin dashed line), while those from lateral/caudal ganglionic eminences (LGE/CGE) (blue) and dorsal origin (purple) survive at similar proportions (thick solid line). Therefore, neocortex (NCx) and corpus callosum (cc) contain comparable density of ₗᴄOLs (blue dots) and dOLs (purple dots) and are devoid of ₘₚOLs (orange dots). (**B**) In the new model, NCx and cc mainly contain dOLs with very low contribution from the ventral origins. ₗᴄOLs mainly contribute to piriform cortex (Pir) and anterior commissure (ac). ₘₚOLs makes a small but sustained contribution to NCx, with a strong laminar preference toward layer 4 in somatosensory cortex (SS). In addition, dOLs and ₘₚOLs also make substantial contributions to Pir and ac, respectively. Gray dots indicate OLs in unanalyzed regions.

cc, ac is mainly populated by $_{LC}$OLs and $_{MP}$OLs and supplemented by very low proportion of dOLs (*Figure 2K–N*).

To substantiate the above results, we further breed *Opalin^{Flp}::Emx1^{Cre}::Nkx2.1^{Cre}*::Ai65 to label dOLs together with $_{MP}$OLs by RFP and co-stained them with ASPA (*Figure 2—figure supplement 3*). RFP−ASPA+ cells were difficult to find in Mo, SS, and cc, but were more easily observed in Pir and ac, consistent with the respective low and high $_{LC}$OL contributions in these regions.

In summary, our findings significantly revised the canonical model of forebrain OL origins (*Figure 3A*), and provided a new and more comprehensive view (*Figure 3B*). We demonstrated that neocortical OLs are mainly derived from dorsal origin with small but lasting contribution from the ventral origin (*Figure 2*, *Figure 2—figure supplements 1B and 2*). Our data showed that LGE/CGE makes little contribution to neocortex and cc, but makes major contribution to piriform cortex and ac (*Figure 2* and *Figure 2—figure supplement 3*). This finding is supported by another report in which in utero electroporation failed to label LGE-derived cortical OLs in both embryonic and early postnatal brains, and an exclusion strategy revealed very low percentage of LGE/CGE-derived cortical OLs in neonatal brains (*Li et al., 2023*). The lack of adult labeling in our study together with the lack of developmental labeling in the other study suggests that the lack of $_{LC}$OL in neocortex is less likely caused by competitive postnatal elimination, but more likely due to limited production and/or allocation. We further discovered that MGE/POA makes a small but persistent contribution to the neocortex with a distinct distribution pattern featured by a rostral-high to caudal-low gradient and a preference toward L4 in SS (*Figure 2—figure supplement 2*). Whether their enduring existence and highly biased localization has functional implications awaits future exploration. In addition, we found that the cc showed a similar OL composition as the neocortex, but the Pir and the ac each exhibited distinct OL compositions in term of their embryonic origins. $_{LC}$OLs are the major contributor to both regions, while dOLs and $_{MP}$OLs mainly contribute to Pir and ac, respectively (*Figure 2*).

In addition to the new framework of forebrain OL origins (*Figure 3*), we also generated a new driver (*Figure 1*) and established multiple combinatorial genetic models (*Figure 2*) for efficient tracking and direct visualization of OLs from different embryonic origins without interference from other cells types sharing the same progenitor domains such as OL precursors, astrocytes, and neurons (*Figures 1–2*). These tools set up a firm foundation and will provide reliable experimental access for future inquiries on the development and function of diverse OLs in healthy and disease brains (*Gong et al., 2022*), especially to uncover the relationship between their developmental origins and the functional and molecular heterogeneity.

# Materials and methods

**Key resources table**

| Reagent type (species) or resource | Designation | Source or reference | Identifiers | Additional information |
|---|---|---|---|---|
| Genetic reagent (*Mus musculus*) | *Nkx2.1^{Cre}* | The Jackson Laboratory | Strain#: 008661; RRID: IMSR_JAX:008661 | |
| Genetic reagent (*Mus musculus*) | *Gsh2^{Cre}* | The Jackson Laboratory | Strain#: 025806; RRID: IMSR_JAX:025806 | |
| Genetic reagent (*Mus musculus*) | *Emx1^{Cre}* | The Jackson Laboratory | Strain#: 005628; RRID: IMSR_JAX:005628 | |
| Genetic reagent (*Mus musculus*) | Ai65 | The Jackson Laboratory | Strain#: 021875; RRID: IMSR_JAX:021875 | |
| Genetic reagent (*Mus musculus*) | RC::FLTG | The Jackson Laboratory | Strain#: 026932; RRID: IMSR_JAX:026932 | |
| Genetic reagent (*Mus musculus*) | Ai62 | The Jackson Laboratory | Strain#: 022731; RRID: IMSR_JAX:022731 | |
| Genetic reagent (*Mus musculus*) | HG-FRT | The Jackson Laboratory | Strain#: 028581; RRID: IMSR_JAX:028581 | |
| Genetic reagent (*Mus musculus*) | *Opalin^{P2A-Flpo-T2A-tTA2}* | This paper | | See Materials and methods, Mice |

*Continued on next page*

*Continued*

| Reagent type (species) or resource | Designation | Source or reference | Identifiers | Additional information |
|---|---|---|---|---|
| Antibody | anti-RFP (goat polyclonal) | SICGEN | Cat# AB0081-200; RRID: AB_2333095 | IF (1:2000) |
| Antibody | anti-RFP (rabbit polyclonal) | Rockland | Cat# 600-401-379; RRID: AB_2209751 | IF (1:2000) |
| Antibody | anti-GFP (chicken polyclonal) | Aves Labs | Cat# GFP-1020; RRID: AB_10000240 | IF (1:1000) |
| Antibody | anti-MBP (rat polyclonal) | AbD Serotec | Cat# MCA409S; RRID: AB_325004 | IF (1:500) |
| Antibody | anti-CC1 (rabbit polyclonal) | Oasis Biofarm | Cat# OB-PRB070; RRID: AB_2934254 | IF (1:500) |
| Antibody | anti-CC1 (mouse polyclonal) | Millipore | Cat# OP80; RRID: AB_2057371 | IF (1:300) |
| Antibody | anti-ASPA (rat polyclonal) | Oasis Biofarm | Cat# OB-PRT005; RRID: AB_2938679 | IF (1:200) |
| Antibody | anti-Sox9 (rabbit polyclonal) | Chemicon | Cat# AB5535; RRID: AB_2239761 | IF (1:2000) |
| Antibody | anti-NeuN (mouse monoclonal) | Millipore | Cat# MAB377; RRID: AB_2298772 | IF (1:500) |
| Sequence-based reagent | Opalin-F | This paper | PCR primers | GGCCTATGTTTGATTTCCAGCACTG |
| Sequence-based reagent | Opalin-R | This paper | PCR primers | AGCACTTATGACTGCTGAGCCGTTC |
| Chemical compound, drug | Tail lysis buffer | Viagen | Cat# 102-T | |
| Chemical compound, drug | Proteinase K | Beyotime | Cat# ST535 | |
| Chemical compound, drug | Sodium pentobarbital | Sigma-Aldrich | Cat# P3761 | |
| Chemical compound, drug | Normal Donkey Serum | Abcam | Cat# ab7475 | |
| Chemical compound, drug | Triton X-100 | Sigma-Aldrich | Cat# X100PC | |
| Chemical compound, drug | Citrate buffer | Oasis-Biofarm | Cat# BR-AB001 | |
| Other | Aqua-mount | Southern Biotech | Cat# 0100-01 | |
| Other | DAPI stain | Invitrogen | Cat# D1306 | (10 mg/ml) |
| Software, algorithm | ImageJ | National Institutes of Health | RRID: SCR_003070 | |
| Software, algorithm | QuPath | Queen's University Belfast | RRID: SCR_018257 | |
| Software, algorithm | Adobe Photoshop | Adobe Systems | RRID: SCR_014199 | |
| Software, algorithm | GraphPad Prism v8.0.1 | GraphPad Software | RRID: SCR_002798 | |

## Mice

All mouse studies were carried out in strict accordance with the guidelines of the Institutional Animal Care and Use Committee of School of Basic Medical Sciences, Fudan University. All husbandry and experimental procedures were reviewed and approved by the same committee (Permit Number: 20210302-137). All applicable institutional and/or national guidelines for the care and use of animals were followed. The following transgenic mouse lines were used in this study: *Nkx2.1*<sup>Cre</sup> (Jax 008661) (*Xu et al., 2008*), *Gsh2*<sup>Cre</sup> (Jax 025806) (*Kessaris et al., 2006*), *Emx1*<sup>Cre</sup> (Jax 005628) (*Gorski et al., 2002*), Ai65 (Jax 021875) (*Madisen et al., 2015*), and RC::FLTG (Jax 026932) (*Plummer et al., 2015*).

The tTA2-dependent tdTomato reporter (TRE-RFP) was derived from Ai62 (Jax 022731) (*Madisen et al., 2015*), by removing LoxP-STOP-LoxP with *E2a-Cre* (Jax 003724). The Flp-dependent H2B-GFP reporter (HG-FRT) was derived from HG-dual (Jax 028581) via removal of loxP flanking STOP cassette by *CMV-Cre* (*He et al., 2016*). The *Opalin^P2A-Flpo-T2A-tTA2* allele was generated by targeted insertion of the T2A-Flpo-P2A-tTA2 sequence immediately before the STOP codon of the endogenous *Opalin* gene using homologous recombination. Gene targeting vector was generated using PCR-based cloning approach as described before (*He et al., 2016*). More specifically, a 4.7-kb 5′ homology arm, a loxP flanking Neo-positive selection cassette, a T2A-Flpo-P2A-tTA2 cassette and a 2.7-kb 3′ homology arm were cloned into a building vector containing the DTA-negative selection cassette to generate the targeting vector. Targeting vector was linearized and transfected into a C57/black6 ES cell line. ES clones that survived through negative and positive selections were first screened by genomic PCR, then confirmed by Southern blotting using appropriate DIG-dUTP-labeled probes. One positive ES cell clone was used for blastocyst injection to obtain male chimera mice carrying the modified allele following standard procedures. Chimera males were bred with C57BL/6J females to confirm germline transmission by genomic PCR. The Neo selection cassette was self-excised during spermatogenesis of F0 chimeras. Heterozygous F1 siblings were bred with one another to establish the colony. Targeting vector construction, ES cell transfections and screening, blastocyst injections, and chimera breeding were performed by Cyagen.

## Genomic PCR

Genomic DNA was prepared from mouse tails. Tissue was lysed by incubation in tail lysis buffer (Viagen, 102-T) with 0.1 mg/ml proteinase K (Diamond, A100706) overnight at 55°C followed by 45 min at 90°C in an air bath to inactivate proteinase K. The lysate was cleared by centrifugation at maximum speed (21,130 G) for 15 min in a table-top centrifuge. Supernatant containing genomic DNA was used as the PCR template for amplifying DNA products. The following primers were used:

*Opalin-F*: 5′-GGCCTATGTTTGATTTCCAGCACTG-3′
*Opalin-R*: 5′-AGCACTTATGACTGCTGAGCCGTTC-3′

## Immunohistochemistry and microscopy

Mice were anesthetized by intraperitoneal injection of 1.5% sodium pentobarbital (0.09 mg/g body weight) and then intracardially perfused with saline followed by 4% paraformaldehyde in 0.1 M phosphate buffer. Following post fixation at 4°C for 24 hr, brain samples were sectioned at 30 µm using a vibratome (Leica VT1000S), or transferred into 30% sucrose in 0.1 M PB for cryoprotection, embedded in optimal cutting temperature (OCT) compound, and sectioned using a cryostat (Leica CM1950). For CC1 immunostaining, antigen retrieval was performed prior to blocking by boiling for 3 min in 10 mM citrate buffer (pH 6.0). Sections were blocked in phosphate buffered saline (PBS) containing 0.05% Triton and 5% normal donkey serum and then incubated with the following primary antibodies in the blocking solution at 4°C overnight: RFP (goat polyclonal antibody, 1:2000, SICGEN AB0081-200; rabbit polyclonal antibody, 1:2000, Rockland 600-401-379), GFP (chicken polyclonal antibody, 1:1000, Aves Labs, GFP-1020), MBP (rat polyclonal antibody, 1:500, AbD Serotec, MCA409S), CC1 (rabbit polyclonal antibody, 1:500, Oasis Biofarm, OB-PRB070, mouse polyclonal antibody, 1:300, Millipore, OP80), ASPA (rat polyclonal antibody, 1:200, Oasis Biofarm, OB-PRT005), Sox9 (rabbit polyclonal antibody, 1:2000, Chemicon, AB5535), and NeuN (mouse monoclonal antibody, 1:500, Millipore, MAB377). Sections were then incubated with appropriate Alexa fluor dye-conjugated IgG secondary antibodies (1:500, Thermo Fisher Scientific) or CF dye-conjugated IgG secondary antibodies (1:250, Sigma) in blocking solution and mounted in Aqua-mount (Southern Biotech, 0100-01). Sections were counterstained with DAPI (4',6-diamidino-2-phenylindole). Sections were imaged with confocal microscopy (Olympus FV3000), fluorescence microscopy (Nikon Eclipse Ni; Olympus VS120; Olympus VS200), and fluorescent stereoscope (Nikon SMZ25). All quantifications were performed in 2-month-old adult mice from coronal sections between Bregma +1.94 and −2.80 mm. Anatomical regions were identified according to the *Paxinos 'The Mouse Brain' Atlas* and the *Allen Reference Atlas*, and their areas were measured in ImageJ for density calculations, whenever applicable. For cortical regions, every fourth section within the range of selection was analyzed. For whiter matter tracts, three consecutive sections at Bregma 0.14 were analyzed. At least three brains were analyzed

for each genotype. To quantify density and co-localization, cells were identified and counted in Adobe Photoshop or ImageJ in conjugation with QuPath.

## Statistical analysis

GraphPad Prism version 8.0.1 was used for statistical calculations. No statistical methods were used to predetermine sample sizes, but our sample sizes are similar to those reported in previous publications. Data collection and analysis were performed blind to the conditions of the experiments whenever possible. No animals or data points were excluded from the analysis. Normalcy was assessed using Shapiro–Wilk test. Equal variances were assessed using $F$ test or Bartlett's test. Statistical significance was tested using two-tailed unpaired $t$-test, Welch's $t$-test, one-way analysis of variance (ANOVA), and two-way ANOVA followed by Tukey's or Bonferroni post hoc test, wherever appropriate. Data are presented as mean ± standard error of the mean. $p < 0.05$ was considered significant. Significance is marked as $*p < 0.05$, $**p < 0.01$, and $***p < 0.001$.

## Acknowledgements

We thank Dr. Yilin Tai for helpful discussion and Dr. Min Jiang from core facility of IOBS, Fudan University, for technical support on imaging experiments. This study was supported by funds from the National Science and Technology Innovation 2030 Major Projects of China (STI2030-Major Projects-2022ZD0206500), National Natural Science Foundation of China (32171087, 32371145).

## Additional information

### Funding

| Funder | Grant reference number | Author |
| --- | --- | --- |
| National Science and Technology Innovation 2030 Major Projects of China | STI2030-Major Projects-2022ZD0206500 | Miao He |
| National Natural Science Foundation of China | 32171087 | Miao He |
| National Natural Science Foundation of China | 32371145 | Ling Gong |

The funders had no role in study design, data collection, and interpretation, or the decision to submit the work for publication.

### Author contributions

Yuqi Cai, Data curation, Formal analysis, Investigation, Visualization, Methodology, Writing - original draft, Writing - review and editing; Zhirong Zhao, Validation, Investigation, Writing - review and editing; Mingyue Shi, Data curation, Validation, Investigation, Writing - review and editing; Mingfang Zheng, Writing - review and editing; Ling Gong, Formal analysis, Funding acquisition, Visualization, Methodology, Writing - original draft, Project administration; Miao He, Conceptualization, Resources, Supervision, Funding acquisition, Visualization, Methodology, Writing - original draft, Writing - review and editing

### Author ORCIDs

Yuqi Cai http://orcid.org/0009-0001-5831-4980
Miao He https://orcid.org/0000-0003-0731-6801

### Ethics

All mouse studies were carried out in strict accordance with the guidelines of the Institutional Animal Care and Use Committee of School of Basic Medical Sciences, Fudan University. All husbandry and experimental procedures were reviewed and approved by the same committee (Permit Number: 20210302-137). All applicable institutional and/or national guidelines for the care and use of animals were followed.

Reviewer #1 (Public review): https://doi.org/10.7554/eLife.95406.3.sa1
Reviewer #2 (Public review): https://doi.org/10.7554/eLife.95406.3.sa2
Reviewer #3 (Public review): https://doi.org/10.7554/eLife.95406.3.sa3
Author response https://doi.org/10.7554/eLife.95406.3.sa4

## Additional files

### Supplementary files
• MDAR checklist

### Data availability
All data generated or analyzed during this study are included in the manuscript. Source data have been provided for Figures 1 and 2.

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
