## [Editor Report · eLife assessment]

In this study the authors revisited the question of the embryonic origin of telencephalic oligodendrocytes using some new and powerful genetic tools. There is **convincing** evidence to support previous suggestions of a predominantly cortical origin of oligodendrocytes in the cerebral cortex, however the new studies suggest that LGE/CGE-derived oligodendrocytes make a modest contribution in some areas, while MGE/POA-derived oligodendrocytes make a small but enduring contribution. The findings are **valuable** and should be of interest to developmental and myelin biologists.

---

## [Referee Report · Reviewer #1 (Public review)]

Summary:

In this study, the authors generated a novel transgenic mouse line OpalinP2A-Flpo-T2A-tTA2 to specifically label mature oligodendrocytes, and at the same time their embryonic origins by crossing with a progenitor cre mouse line. With this clever approach, they found that LGE/CGE-derived OLs make minimum contributions to the neocortex, whereas MGE/POA-derived OLs make a small but lasting contribution to the cortex. These findings are contradictory to the current belief that LGE/CGE-derived OPCs make a sustained contribution to cortical OLs, whereas MGE/POA-derived OPCs are completely eliminated. Thus, this study provides a revised and more comprehensive view on the embryonic origins of cortical oligodendrocytes. To specifically label mature oligodendrocytes, and at the same time their embryonic origins by crossing with a progenitor cre mouse line. With this clever approach, they found that LGE/CGE-derived OLs make minimum contributions to the neocortex, whereas MGE/POA-derived OLs make a small-but-lasting contribution to to cortex. These findings are contradictory to the current belief that LGE/CGE-derived OPCs make a sustained contribution to cortical OLs, whereas MGE/POA-derived OPCs are completely eliminated. Thus, this study has provided a revised and updated view on the embryonic origins of cortical oligodendrocytes.

Strengths:

The authors have generated a novel transgenic mouse line to specifically label mature differentiated oligodendrocytes, which is very useful for tracing the final destiny of mature myelinating oligodendrocytes. Also, the authors carefully compared the distribution of three progenitor cre mouse lines and suggested that Gsh-cre also labeled dorsal OLs, contrary to the previous suggestion that it only marks LGE-derived OPCs. In addition, the author also analyzed the relative contributions of OLs derived from three distinct progenitor domains in other forebrain regions (e.g. Pir, ac). Finally, the new transgenic mouse lines and established multiple combinatorial genetic models will facilitate future investigations of the developmental origins of distinct OL populations and their functional and molecular heterogeneity.

Comments on latest version: In this revised and improved manuscript, the authors have adequately addressed my concerns, and I have no further issues to raise.

---

## [Referee Report · Reviewer #2 (Public review)]

In this manuscript, Cai et al use a combination of mouse transgenic lines to re-examine the question of the embryonic origin of telencephalic oligodendrocytes (OLs). Their tools include a novel Flp mouse for labelling mature oligodendrocytes and a number of pre-existing lines (some previously generated by the last author in Josh Huang's lab) that allowed combinatorial or subtractive labelling of oligodendrocytes with different origins. The conclusion is that cortically-derived OLs are the predominant OL population in the motor and somatosensory cortex and underlying corpus callosum, while the LGE/CGE generates OLs for the piriform cortex and anterior commissure rather than the cerebral cortex. Small numbers of MGE-derived OLs persist long-term in the motor, somatosensory and piriform cortex.

Strengths:

The strength and novelty of the manuscript lie in the elegant tools generated and used. These have enabled the resolution of the issue regarding the contribution of different telencephalic progenitor zones to the cortical oligodendrocyte population.

Comments on latest version:

The revised manuscript by Cai et al has addressed all the issues raised. I have some minor comments:

Figure 2: The y axis in figure 2L should be the same as the y axis in 2M to make the contribution to Mo and SS more clear.

Figure 3: Although this is clear in the figure, A an B should be labelled as classical model and new model to help the reader understand immediately what the two figures show.

Suppl Fig 2: It is not clear what 1-7 represent. It should be made clear in the legend which areas have been pooled into the different bins. The X axis should be labelled.

---

## [Referee Report · Reviewer #3 (Public review)]

In the manuscript entitled "Embryonic Origins of Forebrain Oligodendrocytes Revisited by Combinatorial Genetic Fate Mapping," Cai et al. used an intersectional/subtractional strategy to genetically fate-map the oligodendrocyte populations (OLs) generated from medial ganglionic eminence (NKX2.1+), lateral ganglionic eminences, and dorsal progenitor cells (EMX1+). Specifically, they generated an OL-expressing reporter mouse line OpalinP2A-Flpo-T2A-tTA2 and bred with region-specific neural progenitor-expressing Cre lines EMX1-Cre for dOL and NKX2.1-Cre for MPOL. They used a subtractional strategy in the OpalinFlp::Emx1Cre::Nkx2.1Cre::RC::FLTG mouse line to predict the origins of OLs from lateral/caudal ganglionic eminences (LC). With their genetic tools, the authors concluded that neocortical OLs primarily consist of dOLs. Although the populations of OLs (dOLs or MP-OLs) from Emx1+ or Nkx2.1+ progenitors are largely consistent with previous findings, they observed that MP-OLs contribute minimally but persist into adulthood without elimination as in the previous report (PMID: 16388308).

Intriguingly, by using an indirect subtraction approach, they hypothesize that both Emx1-negative and Nkx2.1-negative cells represent the progenitors from lateral/caudal ganglionic eminences (LC), and conclude that neocortical OLs are not derived from the LC region. This is in contrast to the previous observation for the contribution of LC-expressing progenitors (marked by Gsx2-Cre) to neocortical OLs (PMID: 16388308). The authors claim that Gsh2 is not exclusive to progenitor cells in the LC region (PMID: 32234482). However, Gsh2 exhibits high enrichment in the LC during early embryonic development. The presence of a small population of Gsh2-positive cells in the late embryonic cortex could originate/migrate from Gsh2-positive cells in the LC at earlier stages (PMID: 32234482). Consequently, the possibility that cortical OLs derived from Gsh2+ progenitors in LC could not be conclusively ruled out. Notably, a population of OLs migrating from the ventral to the dorsal cortical region was detected after eliminating dorsal progenitor-derived OLs (PMID: 16436615).

The indirect subtraction data for LC progenitors drawn from the OpalinFlp-tdTOM reporter in Emx1-negative and Nkx2.1-negative cells in the OpalinFlp::Emx1Cre::Nkx2.1Cre::RC::FLTG mouse line present some caveats that could influence their conclusion. The extent of activity from the two Cre lines in the OpalinFlp::Emx1Cre::Nkx2.1Cre::RC::FLTG mice remains uncertain. The OpalinFlp-tdTOM expression could occur in the presence of either Emx1Cre or Nkx2.1Cre, raising questions about the contribution of the individual Cre lines. To clarify, the authors should compare the tdTOM expression from each individual Cre line, OpalinFlp::Emx1Cre::RC::FLTG or OpalinFlp::Nkx2.1Cre::RC::FLTG, with the combined OpalinFlp::Emx1Cre::Nkx2.1Cre::RC::FLTG mouse line. This comparison is crucial as the results from the combined Cre lines could appear similar to only one Cre line active.

Overall, the authors provided intriguing findings regarding the origin and fate of oligodendrocytes from different progenitor cells in embryonic brain regions. However, further analysis is necessary to substantiate their conclusion about the fate of LC-derived OLs convincingly.

Comments on latest version: The overall responses by the authors are satisfactory.

---

## [Author Response]

The following is the authors’ response to the original reviews.

**Reviewer #1 (Public Review):**
[...] Strengths:The authors have generated a novel transgenic mouse line to specifically label mature differentiated oligodendrocytes, which is very useful for tracing the final destiny of mature myelinating oligodendrocytes. Also, the authors carefully compared the distribution of three progenitor cre mouse lines and suggested that Gsh-cre also labeled dorsal OLs, contrary to the previous suggestion that it only marks LGE-derived OPCs. In addition, the author also analyzed the relative contributions of OLs derived from three distinct progenitor domains in other forebrain regions (e.g. Pir, ac). Finally, the new transgenic mouse lines and established multiple combinatorial genetic models will facilitate future investigations of the developmental origins of distinct OL populations and their functional and molecular heterogeneity.Weaknesses:Since OpalinP2A-Flpo-T2A-tTA2 only labels mature oligodendrocytes but not OPCs, the authors can not suggest that the lack of LGE/CGE-derived-OLs in the neocortex is less likely caused by competitive postnatal elimination, but more likely due to limited production and/or allocation (line 118-9). It remains possible that LGE/CGE-derived OPCs migrate into the cortex but are later eliminated.

We are glad that the reviewer appreciates our work and are grateful for the positive comments and the constructive suggestion. We agree with the reviewer that our methodology by itself cannot suggest whether the lack of LGE/CGE-derived-OLs in the neocortex is caused by competitive postnatal elimination or not. That is why we cited a parallel work by Li et al. (ref [17] in the original manuscript; ref [19] in the revised manuscript), in which in utero electroporation (IUE) failed to label LGE-derived OL lineage cells in both embryonic and early postnatal brains. Although they did not directly explore CGE using IUE, their fate mapping results using Emx1-Cre; Nkx2.1-Cre; H2B-GFP at P0 and P10 revealed very low percentage of LGE/CGE-derived OL lineage cells. The lack of adult labeling in our study together with the lack of developmental labeling in the other study prompted us to hypothesize that the lack of LGE/CGE-derived-OLs in the neocortex is less likely caused by competitive postnatal elimination, but more likely due to limited production and/or allocation. In the revised manuscript, we have expanded the discussion to explain this point more clearly.

**Reviewer #2 (Public Review):**
[...] Strengths:The strength and novelty of the manuscript lies in the elegant tools generated and used and which have the potential to elegantly and accurately resolve the issue of the contribution of different progenitor zones to telencephalic regions.

We are glad that the reviewer appreciates our work and are grateful for the overall positive comments.

Weaknesses:(1) Throughout the manuscript (with one exception, lines 76-78), the authors quantified OL densities instead of contributions to the total OL population (as a % of ASPA for example). This means that the reader is left with only a rough estimation of the different contributions.

We thank the reviewer for this constructive suggestion. We have replaced the density quantification (Figure 2F and 3D in the original manuscript) with contributions to the total OL population (% of ASPA) (Figure 2J and 2N in the revised manuscript).

(2) All images and quantifications have been confined to one level of the cortex and the potential of the MGE and the LGE/CGE to produce oligodendrocytes for more anterior and more posterior cortical regions remains unexplored.

The quantifications were not confined to one level of the cortex but were performed in brain sections ranging from Bregma +1.94 to -2.80 mm, as shown in Supplementary Figure 2A-B in the original manuscript. We apologize for not having stated and presented this information clearly enough, and for the confusions it may have caused. In the revised manuscript, we have added relevant descriptions in the “Material and Methods” section (line 199-200*) and schematics along with representative images of more anterior and more posterior cortical regions (Supplementary Figure 2A-D).

(3) Hence, the statement that "In summary, our findings significantly revised the canonical model of forebrain OL origins (Figure 4A) and provided a new and more comprehensive view (Figure 4B)." (lines 111, 112) is not really accurate as the findings are neither new nor comprehensive. Published manuscripts have already shown that (a) cortical OLs are mostly generated from the cortex [Tripathi et al 2011 (https://doi.org/10.1523/JNEUROSCI.6474-10.2011), Winker et al 2018 (https://doi.org/10.1523/JNEUROSCI.3392-17.2018) and Li et al (https://doi.org/10.1101/2023.12.01.569674)] and (b) MGE-derived OLs persist in the cortex [Orduz et al 2019 (https://doi.org/10.1038/s41467-019-11904-4) and Li et al 2024 (https://doi.org/10.1101/2023.12.01.569674)]. Extending the current study to different rostro-caudal regions of the cortex would greatly improve the manuscript.

As explained in the response to comment (2), our original quantifications included different rostro-caudal regions of the cortex. In the revised manuscript, we have added more schematics and representative images in the Supplementary Figure 2 for better illustration to resolve the concern of comprehensiveness.

We thank the reviewer for listing and summarizing highly relevant published researches along with the parallel study by Li et al. submitted to eLife. We apologize for the omission of the first two references in our original manuscripts and have cited them in appropriate places (ref [10] and ref [11] in the revised manuscript). However, we believe these works do not compromise the novelty and significance of our work for the following reasons:

(1) Tripathi et al. 2011 (ref [10] in the revised manuscript) analyzed OL lineage cells in the corpus callosum and the spinal cord, but not in the cortex and anterior commissure. Their analysis was performed in juvenile mice (P12/13), not in adulthood. Most importantly, their analysis of ventrally derived OL lineage cells relied on lineage tracing using Gsh2Cre, which in fact also label OLs derived from Gsh2+ dorsal progenitors. In contrast, we analyzed mature OLs in the cortex, corpus callosum and anterior commissure in 2-month-old adult mice. We used intersectional and subtractive strategy to label OLs derived from dorsal, LGE/CGE and MGE/POA origins. Our strategy differentiated the two different ventral lineages (LGE/CGE vs. MGE/POA) and avoided mixed labeling of OLs from ventral and dorsal Gsh2+ progenitors.

(2) Winkler et al. 2018 (ref [11] in the revised manuscript) analyzed OLs derived from dorsal progenitors but only quantified those in the gray matter and the white matter of somatosensory cortex. Their quantification relied on co-staining with Olig2/Sox10, and thereby included both oligodendrocyte precursors (OPCs) and OLs. In contrast, we analyzed mature OLs from three origins and quantified not only neocortical regions (Mo and SS) but also an archicortical region (Pir). Our analysis revealed that although dorsally derived OLs dominate neocortex, ventrally derived OLs, especially the LGE/CGE-derived ones, dominate piriform cortex.

(3) Orduz et al. 2019 (ref [7] in the original manuscript and the revised manuscript) mainly focused on POA-derived OLs in the somatosensory cortex. Although they performed limited analysis on MGE/POA-derived OPCs at postnatal day 10 and 19, no quantification of MGE/POA-derived OLs was performed in terms of their density, contribution to the total OL population and spatial distribution in the cortex. In contrast, we performed systematic quantification on these aspects to demonstrate that MGE/POA-derived OLs make small but sustained contribution to cortex with a distribution pattern distinctive from those derived from the dorsal origin.

(4) Li et al. 2024 (ref [17] in the original manuscript and [19] in the revised manuscript) is a parallel study submitted to eLife. Their and our independent discoveries nicely complemented each other. Using different sets of techniques and experiments but some shared genetic mouse models, we both found that LGE/CGE made minimum contribution to neocortical OLs. Their analysis in the prenatal and early postnatal stages together with our analysis in the adult brain painted a more comprehensive picture of cortical oligodendrogenesis. The uniqueness of our work is that we performed systematic quantification of all three origins and uncovered the differential contributions to neocortex, piriform cortex, corpus callosum and anterior commissure.

In summary, our work developed novel strategies to faithfully trace OLs from the three different origins and performed systematic analysis in the adult brain. Our data uncovered their differential contributions to neocortex, piriform cortex and the two commissural white matter tracts, which significantly differ not only from the canonical view but also from other previous studies in aspects discussed above. We believe our discoveries did significantly revise the canonical model of forebrain OL origins and provided a new and more comprehensive view.

**Reviewer #3 (Public Review):**
[...] Intriguingly, by using an indirect subtraction approach, they hypothesize that both Emx1-negative and Nkx2.1-negative cells represent the progenitors from lateral/caudal ganglionic eminences (LC), and conclude that neocortical OLs are not derived from the LC region.The authors claim that Gsh2 is not exclusive to progenitor cells in the LC region (PMID: 32234482). However, Gsh2 exhibits high enrichment in the LC during early embryonic development. The presence of a small population of Gsh2-positive cells in the late embryonic cortex could originate/migrate from Gsh2-positive cells in the LC at earlier stages (PMID: 32234482). Consequently, the possibility that cortical OLs derived from Gsh2+ progenitors in LC could not be conclusively ruled out. Notably, a population of OLs migrating from the ventral to the dorsal cortical region was detected after eliminating dorsal progenitor-derived OLs (PMID: 16436615).The indirect subtraction data for LC progenitors drawn from the OpalinFlp-tdTOM reporter in Emx1-negative and Nkx2.1-negative cells in the OpalinFlp::Emx1Cre::Nkx2.1Cre::RC::FLTG mouse line present some caveats that could influence their conclusion. The extent of activity from the two Cre lines in the OpalinFlp::Emx1Cre::Nkx2.1Cre::RC::FLTG mice remains uncertain. The OpalinFlp-tdTOM expression could occur in the presence of either Emx1Cre or Nkx2.1Cre, raising questions about the contribution of the individual Cre lines. To clarify, the authors should compare the tdTOM expression from each individual Cre line, OpalinFlp::Emx1Cre::RC::FLTG or OpalinFlp::Nkx2.1Cre::RC::FLTG, with the combined OpalinFlp::Emx1Cre::Nkx2.1Cre::RC::FLTG mouse line. This comparison is crucial as the results from the combined Cre lines could appear similar to only one Cre line active.Overall, the authors provided intriguing findings regarding the origin and fate of oligodendrocytes from different progenitor cells in embryonic brain regions. However, further analysis is necessary to substantiate their conclusion about the fate of LC-derived OLs convincingly.

We thank the reviewer for these thoughtful comments. We agree with the reviewer that the presence of Gsh2-positive cells in the late embryonic cortex by itself could not rule out the possibility that they originate/migrate from Gsh2-positive cells in the LC at earlier stages. Staining dorsal-lineage intermediate progenitors with Gsh2, or performing intersectional lineage tracing using Gsh2Cre along with a dorsal-specific Flp driver, would provide more direct evidence on this issue. Nonetheless, as our lineage tracing of LGE/CGE-derive OLs did not employ Gsh2Cre, the doubt on the identity of Gsh2+ cortical progenitors should not affect the interpretation of our data.

Regarding the subtractional LCOL labeling strategy used in our study, we wonder if there was any misunderstanding by the reviewer. As stated in our manuscript (line 59-61) and reiterated by the reviewer, OpalinFlp::Emx1Cre::Nkx2.1Cre::RC::FLTG labels OLs derived from progenitors that express neither Emx1Cre nor Nkx2.1Cre. As these two progenitor pools do not overlap with each other, there is a purely additive effect of their actions. If there is any concern about efficiency and specificity, it would be non-adequate Cre-mediated recombinations that lead to mislabeling of dOLs or MPOLs as LCOLs (i.e., OLs derived from Emx1 or Nkx2.1-expressing progenitors were not successfully “subtracted” and thereby “wrongly” retained RFP expression). Therefore, the bona-fide LGE/CGE-derive OLs would only be fewer but not more than RFP+ LCOLs labeled by our subtractional strategy, even if any of the Cre lines did not work efficiently enough. In any case, this would not affect our conclusion that LGE/CGE-derive OLs make a minimal contribution to neocortex, as the “ground truth” contribution by LGE/CGE could only be less but not more than what we have observed using the current strategy.

In support of our conclusion, a parallel study by Li et al. 2024 (ref [17] in the original manuscript; ref [19] in the revised manuscript) also provided independent experimental evidence that “any contribution of oligodendrocyte precursors to the developing cortex from the lateral ganglionic eminence is minimal in scope (quoted from its eLife assessment).” In addition, in their revision, they performed Gsh2 immunostaining in P0 Emx1Cre::HG-loxP mouse and found nearly all Gsh2+ cells in the cortical SVZ were derived from the Emx1+ lineage. We are glad that this additional piece of evidence further clarified the case, but still want to emphasize that the subtractional strategy we took was designed purposefully to avoid the potential uncertainty of Gsh2Cre and to more faithfully label LGE/CGE-derived OLs. Therefore, the validity of our conclusion about the fate of LC-derived OLs should be independent from the question on the identity of Gsh2+ cortical progenitors and stands well by itself.

We hope that these explanations have adequately addressed the reviewer’s concerns.

**Recommendations for the authors:**

**Reviewer #2 (Recommendations For The Authors):**
In Figures 2C, 2D, 2E and 3D, the authors should provide counts of labelled cells as a % of ASPA+ cells. This will give an accurate picture of the contribution of the different progenitor regions to OLs.The graphs in Figure 2F are unnecessary since they are simply repeats of C-E but re-arranged.

We thank the reviewer for the valuable suggestions. These two recommendations are sort of related, and thereby we made the following changes. We replaced the density quantification in Figure 2F and 3D with % of ASPA (Figure 2J and 2N in the revised manuscript) to give an accurate picture of the contribution of the different progenitor regions to OLs, as suggested by the reviewer. We still retained the density counts in Figure 2C-E (Figure 2G-I in the revised manuscript). Together with quantifications of rotral-caudal and larminar distributions presented in Supplementary Figure 2, these data demonstrated that OLs from differential origins display distinct spatial distribution patterns.

At what ages were the quantifications performed in all the figures?

We apologize for the omission of this information in the original manuscript. All quantifications were performed in 2-month-old adult mice. We have added this information in the “Material and Methods” section of the revised manuscript.

In 2D, and 3B the GFP should have been activated but the authors do not show it or quantify it presumably because GFP would flood the sections in the presence of Emx1Cre. Nevertheless, since eGFP is shown in the diagram in 2B, the authors should mention why they chose not to show it.

We thank the reviewer for the helpful comment and the suggestion. We have modified the schematic in Figure 2B and added explanation in the figure legend (line 308-313). We also added a schematic in Supplementary Figure 1A along with images of GFP channel in Supplementary Figure 1D (line 338-350).

All the main figures and supplementary figures are too small to see properly.

We are sorry that there was severe compression of images in the combined manuscript file at the conversion step during the initial submission. We apologize for the compromised image quality and have re-uploaded full-size figures as individual files on BioRxiv soon after receiving the reviews. For the revised manuscript, we also take care to upload full-size figures at high resolution as individual files to ensure their quality of presentation.

Supplementary Figure 2E is unnecessary and perhaps misleading the reader that cortical-derived OLs have a preference for the lower layers whereas the distribution may simply reflect the distribution of OLs in the cortex.

We thank the reviewer for the helpful comment and the suggestion. We have removed this panel and replaced it with quantifications of relative laminar distributions of the total (ASPA+) OLs along with those from the three different origins (Supplementary Figure 2G in the revised manuscript). Indeed, the preference for the lower layers of dorsally-derived OLs mirrored the distribution of total OLs in the cortex, while the MGE/POA-derived OLs deviate significantly from others and exhibit higher preference towards layer 4.

Quantification of labelled cells as a % of ASPA should also be performed in Supplementary Figure 3.

We thank the reviewer for this suggestion. In the revised manuscript, we have included quantifications of labelled cells as % of ASPA for both OpalinFlp::Emx1Cre::Ai65 and OpalinFlp::Nkx2.1Cre::Ai65 (Figure 2J and N). The sum of the these two data sets will be equivalent to those of OpalinFlp::Emx1Cre::Nkx2.1Cre::Ai65 shown in Supplementary Figure 3, and thereby we did not perform additional quantifications to avoid redundant efforts.

Imaging and quantification should be extended to more posterior regions of the cortex to find out whether the contribution is different from the areas already examined.

We thank the reviewer for the suggestion on imaging and apologize for the confusion about the range of quantification. As explained in the response to comment (2) of weakness, the quantifications were not confined to one level of the cortex but were performed in brain sections ranging from Bregma +1.94 to -2.80 mm, as shown in Supplementary Figure 2A-B in the original manuscript. In the revised manuscript, we have added relevant descriptions in the “Material and Methods” section (line 199-200) and schematics along with representative images of more anterior and more posterior cortical regions (Supplementary Figure 2A-D).

**Reviewer #3 (Recommendations For The Authors):**
(1) The authors should provide Opalin reporter expression data across various brain regions at different developmental stages to clarify the expression pattern of the reporter.

We appreciate the reviewer’s comment. We chose to performed all quantifications in adult mice as Opalin is a well-established marker for differentiated OLs and the recombinase-dependent reporter expression is accumulative and irreversible. If there is any non-specific labeling in any earlier developmental stage, it would be retained and manifested at the timepoint we examined as well. In another word, the fact that we did not detect any non-specific labeling in the current dataset but only confined labeling in mature OLs ensured that no non-OL labeling was present in earlier timepoint. As shown in Figure 1D-F, reporter expression activated by the Opalin driver is presented at high OL specificity in all analyzed brain regions. This is further corroborated by results from combinatorically labeled samples (Figure 2 and Supplementary Figure 2), in which only OLs but not any other cell types were labeled in all analyzed brain regions too. Following the reviewers’ suggestions, we have added representative images of more rostral and more caudal cortical regions (Supplementary Figure 2B-D), which also showed highly specific OL labeling.

(2) In Figure 1D, please specify the developmental stage of the mice used for staining.

We apologize for the omission of this information in the original manuscript. All quantifications were performed in 2-month-old adult mice. We have added this information in the “Material and Methods” section (line 199-200) of the revised manuscript.

(3) The authors should clarify if the Opalin reporter expressed in OPCs and astrocytes at developmental stages of mice, such as P0, P7, and P30.

We appreciate the reviewer’s comment, but as explained in response to comment (1), Opalin is a well-established marker for differentiated OLs which is not expressed in OPCs or astrocytes. As shown in Figure 1D-E, reporter expression is confined to CC1+ differentiated OLs with no colocalization with Sox9 (astrocyte marker). In support with this observation, only ASPA+ differentiated OLs but no OPC or astrocyte were labeled in any of the combinatorial lineage tracing samples generated using this line combined with progenitor-Cre lines. In addition to marker staining, we also did not observe any RFP+ cells with OPC or astrocyte morphology. As the recombinase-dependent reporter expression is accumulative and irreversible, the fact no non-specific labeling was observed in adult brain retrospectively proved the specificity of Oplain-Flp in earlier developmental stages.

(4) In Figure 1E, authors should address why the efficiency of the tdTomato line is notably lower compared to that of H2B-GFP and whether the stability of reporters could impact the conclusions drawn.

The difference in reporting efficiency is mainly caused by differences inherent to the two reporting systems. The TRE-RFP reporter is derived from Ai62, composed of a Tet response element and tdTomato inserted into the T1 TIGRE locus. The tdTomato expression is driven by tTA-TRE transcriptional activation. The HG-loxP reporter is derived from HG-Dual, composed of a CAG promoter, a frt-flanked STOP cassette, and H2B-GFP inserted into the Rosa26 locus. The H2B-GFP expression is driven by CAG promoter after Flp-mediated removal of the STOP cassette. A Flp-dependent tdTomato reporter designed in the same way as the HG-FRT reporter would have similar efficiency. In fact, the RC::FLTG reporter can be viewed as such a reporter in the absence of Cre, which did show similarly high efficiency as HG-FRT and supported efficient subtractive labeling of LGE/CGE-derived OLs. We apologize for a typo in the title of the Y-axis of the right panel in the original Figure 1F which may have caused potential misunderstanding. The “RFP+CC1+/CC1” should be “XFP+CC1/CC1”. We have corrected this mistake and revised the figure legend for clearer description of the data (Line 293-302 in the revised manuscript).

(5) In Figure 2, please clarify the developmental stage of the mice used for staining. Authors should present the eGFP image in addition to tdTOM.

We apologize for the omission of the age information in the original manuscript. All quantifications were performed in 2-month-old adult mice. We have added this information in the “Material and Methods” section (line 199-200) of the revised manuscript. We thank the reviewer for the suggestion on eGFP image and have presented it in supplementary Figure 1 in the revised manuscript.

(6) in Figure 2D, authors should display the eGFP image alongside the tdTomato image. It is difficult to assess the efficiency of Emx-Cre and Nkx2.1-Cre.

We thank the reviewer for the suggestion on eGFP image and have presented eGFP image in Supplementary Figure 1D in the revised manuscript. There are two reasons why we chose to present it in the supplementary figure instead of main figure. First, we added ASPA staining in the green channel along with quantifications of RFP cells as % of ASPA in Figure 2 in the revised manuscript, following reviewer #2’s suggestion. Second, as pointed out by reviewer #2, GFP would flood the sections in the presence of Emx1Cre and could be quite distractive if it was shown together with RFP.

We were not entirely sure what exactly the reviewer means by “assess the efficiency of Emx-Cre and Nkx2.1-Cre”, but we believe that the quantifications of RFP cells as % of ASPA clarified the contribution of each origin to the total OLs (Figure 2J and 2N in the revised manuscript).

(7) Figure 3 depicts the entire brain, replicating the image presented in Figure 2. It would be beneficial to consolidate Figures 2 and 3, as they showcase identical brain scans of different regions.

We thank the reviewer for the constructive suggestion and have consolidated Figures 2 and 3 in the original manuscript into Figure 2 in the revised manuscript.